A deep learning method for drug-target affinity prediction based on sequence interaction information mining

Jiang Mingjian 1
Shao Yunchang 1 sycofqut@163.com
Zhang Yuanyuan 1
Zhou Wei 1
Pang Shunpeng 2
1 School of Information and Control Engineering, Qingdao University of Technology , Qingdao, Shandong , China
2 School of Computer Engineering, WeiFang University , Weifang, Shandong , China
Gillespie Joseph
Electronic publication date: 2023 Dec 11
Publication date: 2023
Volume: 11
Electronic Location ID: e16625
Received 2023 Aug 9; Accepted 2023 Nov 16
Copyright: © 2023 Jiang et al.
Copyright year: 2023
Copyright holder: Jiang et al.
License: This is an open access article distributed under the terms of the Creative Commons Attribution License, which permits unrestricted use, distribution, reproduction and adaptation in any medium and for any purpose provided that it is properly attributed. For attribution, the original author(s), title, publication source (PeerJ) and either DOI or URL of the article must be cited.
License URL: https://creativecommons.org/licenses/by/4.0/

Keywords: Deep learning, Drug-target affinity prediction, Protein sequence, Graph neural network, Convolutional neural network

Funding: National Natural Science Foundation of Shandong Province ZR2022QF111 This work was supported by the National Natural Science Foundation of Shandong Province (No. ZR2022QF111). The funders had no role in study design, data collection and analysis, decision to publish, or preparation of the manuscript.

==============================
Background

A critical aspect of in silico drug discovery involves the prediction of drug-target affinity (DTA). Conducting wet lab experiments to determine affinity is both expensive and time-consuming, making it necessary to find alternative approaches. In recent years, deep learning has emerged as a promising technique for DTA prediction, leveraging the substantial computational power of modern computers.

Methods

We proposed a novel sequence-based approach, named KC-DTA, for predicting drug-target affinity (DTA). In this approach, we converted the target sequence into two distinct matrices, while representing the molecule compound as a graph. The proposed method utilized k-mers analysis and Cartesian product calculation to capture the interactions and evolutionary information among various residues, enabling the creation of the two matrices for target sequence. For molecule, it was represented by constructing a molecular graph where atoms serve as nodes and chemical bonds serve as edges. Subsequently, the obtained target matrices and molecule graph were utilized as inputs for convolutional neural networks (CNNs) and graph neural networks (GNNs) to extract hidden features, which were further used for the prediction of binding affinity.

Results

In order to evaluate the effectiveness of the proposed method, we conducted several experiments and made a comprehensive comparison with the state-of-the-art approaches using multiple evaluation metrics. The results of our experiments demonstrated that the KC-DTA method achieves high performance in predicting drug-target affinity (DTA). The findings of this research underscore the significance of the KC-DTA method as a valuable tool in the field of in silico drug discovery, offering promising opportunities for accelerating the drug development process. All the data and code are available for access on https://github.com/syc2017/KCDTA.

Introduction

Molecule screening plays a pivotal role in drug design by identifying small molecules that have the potential to bind to target protein. However, traditional laboratory experiments for molecule screening are expensive and time-consuming. To overcome these limitations, computer-aided techniques, collectively known as virtual screening techniques, have emeraged. One such technique is molecular docking (Li, Fu & Zhang, 2019). Molecular docking employs a search algorithm to predict the three-dimensional complexes formed by the target protein and its ligand. The search algorithm explores the conformational space to identify the most favorable binding orientations and conformations of the ligand within the protein’s binding site. Subsequently, a scoring function is applied to rank the predicted complex structures based on the calculated binding energy. These methods involve the calculation of structural information, thus categorizing them as structure-based methods. However, structure-based methods have limitations as there are many proteins without structures. To overcome the limitations of structure-based methods, sequence-based methods have emerged as a solution. In sequence-based approaches, many methods use SMILES (Weininger, 1988) to represent small molecules. The full name of “SMILES” is “Simplified Molecular Input Line Entry System.” It is a standardized approach that uses ASCII strings to explicitly describe molecular structures, enabling the conversion of small molecule structures into corresponding string representations. Sequence-based methods require further representation and feature extraction for proteins and small molecules. Li et al. (2022b) has introduced a wide range of feature representations for small molecules. Pahikkala et al. (2015) utilizes target-target and drug-drug similarity as features and employs the Kronecker regularized least squares (KronRLS) algorithm to predict drug-target affinity. It effectively captures the relationships between targets and drugs, enabling accurate affinity predictions. Another method, SimBoost (He et al., 2017), employs a simulation-based approach to train a gradient boosting machine. SimBoost introduces three novel features that capture the nonlinear relationship between drug-target features and affinities. The rapid proliferation of deep learning in diverse domains, including image classification, natural language processing, and speech recognition, has sparked a surge in its utilization for analyzing biological data as well. As an example, DeepDTA (Öztürk, Özgür & Ozkirimli, 2018) employs an embedding layer to encode molecule SMILES and protein sequence. The embeddings are then fed into two separate convolutional networks to extract relevant features, and the extracted features are integrated to achieve the affinity prediction. WideDTA (Öztürk, Ozkirimli & Özgür, 2019) further improves DeepDTA, it enhances the performance by incorporating two additional features for the small molecule and protein. GraphDTA (Nguyen et al., 2021) takes the approach of representing small molecules as graphs and utilizes graph neural networks for feature extraction, thus providing more structural information for affinity mining. MDeePred (Rifaioglu et al., 2021), on the other hand, generates digital matrices that represent the physical, chemical and biological properties of protein sequences, and ECFP4 fingerprint (Rogers & Hahn, 2010) is used for molecule representation. MGraphDTA (Yang et al., 2022) employs a deep graph convolutional neural network to capture the chemical structure information of molecules, which enables the extraction of key structural features for effective affinity prediction. MPS2IT-DTI (De Souza, Fernandes & de Melo Barbosa, 2022) processes protein sequence and molecule using k-mers segmentation (Compeau, Pevzner & Tesler, 2011; Melsted & Pritchard, 2011; Rizk, Lavenier & Chikhi, 2013), and subsequently converts them into images using feature frequency profiles (FFP) (Sims et al., 2009) to represent protein and molecule, which reflects the advantage of k-mers segmentation expression. BACPI (Li et al., 2022a) is an attention-based approach. Processed representations of small molecules and proteins are separately fed into graph neural networks and convolutional neural networks to extract features. The obtained features are then input into a bidirectional attention neural network, which includes attention mechanisms for atoms to protein and amino acids to compound, allowing for a more accurate capture of local effective points for atoms and amino acids. SimCNN (Shim et al., 2021) is a similarity-based approach that utilizes convolutional neural networks to extract features from the outer product of column vectors representing drugs and targets similarity matrices, for the prediction of drug-target affinity. NerLTR-DTA (Ru et al., 2022) extracts features based on the similarity and sharing of neighboring drugs (or proteins) and uses these features as input for learning to rank (LTR) algorithm. Through the ranking framework, it can predict the priority order of affinity between query drugs (proteins) and target proteins (drugs).

In some sequence-based approaches such like GraphDTA and DeepDTA, protein sequences need to be truncated due to limitations in neural network inputs, resulting in the loss of crucial information on long sequences. In this article, we proposed KC-DTA, a sequence-based method that ensures the integrity of protein sequences for drug-target affinity prediction. The protein sequences are mapped into matrices using the k-mers and Cartesian product calculation to capture the hidden information of interactions among residues, which offers a novel approach that could comprehensively represent protein sequences. The proposed method was evaluated on several benchmarks using various performance measures, and the results indicate that the KC-DTA has a high performance on drug-target affinity prediction.

Materials and Methods

Our approach, called KC-DTA, involves the mapping of the protein sequence into two matrices using k-mers segmentation analysis and Cartesian product calculation. In the prediction of drug-target binding affinity, it is common to consider the interaction between small molecules and proteins. This interaction typically occurs in a specific region of the protein, which is represented as a continuous segment in the protein’s amino acid sequence. To acquire protein information more accurately, we employ the k-mers method, which captures local segments of the protein, aiding in a deeper understanding of its structure and function. Furthermore, we also utilize the Cartesian product method to account for interactions between amino acids that are farther apart in the protein sequence. Through the Cartesian product, we can obtain combinations of all amino acids in the protein, facilitating the identification of effective positions within the protein and thus a more comprehensive understanding of its function and interactions. Combining the k-mers method with the Cartesian product method and considering local segments and all combinations of amino acids allows for a more comprehensive representation of protein sequence features. In addition, we represent the molecule as a graph. The resulting protein matrices and molecule graph are then fed into neural networks for feature extraction. The complete model architecture is illustrated in Fig. 1.

Figure 1 This figure illustrates the conversion of protein sequences into two-dimensional and three-dimensional matrix representations, as well as the conversion of small molecule SMILES into graphs.

These feature representations are then separately fed into their respective neural networks to extract features and predict affinity values.

Protein and molecule representation

The protein sequence is represented by two ways and two corresponding matrices are generated. The first way to represent sequence is to count the occurrences of k-mers segmentation for the target protein. For instance, assuming the residue symbol set of protein is P = {A, C, D, E, F, G, H, I, K, L, M, N, P, Q, R, S, T, V, W, X, Y}, which each element of P represents single-letter abbreviation of the residue (amino acid). Next, the combinations of all three-residue are listed to form a combination set Paaa = {AAA, AAC, …YYY}. And supposing the sequence of target protein S is “ACDAG”, then the sequence is first applied by the k-mers (k = 3) segmentation with a step of 1,which results in a set Ks = {ACD, CDA, DAG} for the target sequence S. Finally, the occurrences of the segmentations in Ks is counted regardless of the order of the residues, which means that “ACD” is considered equivalent to “ADC”, “ACD”, “CDA”, “CAD”, “DAC” and “DCA”, and the results are mapped to the above combination set Paaa, thus, a three-dimension matrix OsK with a shape of L×L×L is established, which L is the length of P and the value of L×L×L is the shape of Paaa. For the target protein in the example, the statistical results for “ADC”, “ACD”, “CDA”, “CAD”, “DAC” and “DCA” are all the same, with the k-mers analysis showing a value of 2 for each of them. Because the construction of the matrix ignores the order of three-residue combination, the resulting matrix OsK is a symmetric matrix. The detailed is illustrated in Fig. 2.

Figure 2 This figure illustrates the process of transforming protein sequence into a three-dimensional matrix using the k-mers operation.

The second way to represent protein sequence is to transform sequence into another matrix using Cartesian product calculation operation. In this approach, we employ the Cartesian product to process protein sequence, which all possible combinations of two residues in a protein sequence are represented, enabling a more accurate capture of the protein features. For example, similar to the first way, at first, a set of symbols that can represent all possible protein sequences is denoted as P = {A, C, D, E, F, G, H, I, K, L, M, N, P, Q, R, S, T, V, W, X, Y}. But when form the full combination set, it is different from the first way, which the set is established by listing all two-residue combinations and result in a set Paa = {AA, AC, AD, … YY}. Then the set of all residues of the target sequence is processed using a Cartesian product operation with itself, which the Cartesian product operation on sequence S is defined as Cq = S×S = {<a, b>; a, b ∈ S}, where a, b are residues belong to sequences S. More specifically, if the sequence of the target protein S is “ACDAG”, the Cartesian product operation on the sequence S could yield a set of CS = {AA, AC, AD, AA, AG, CA, CC, CD, CA, CG, DA, DC, DD, DA, DG, AA, AC, AD, AA, AG, GA, GC, GA, GA, GG}. After obtaining the set CS, the occurrences of all two-residue combinations in it are counted with the order of the residues and further mapped to Paa, which results in a two-dimension occurrence matrix OsC with a shape of L×L. Different from the first way of matrix construction, this matrix takes into account the order of two-residue combination, resulting in a non-symmetric matrix. and the processing is shown in Fig. 3.

Figure 3 This figure illustrates the process of transforming protein sequence into a two-dimensional matrix using the Cartesian product operation.

After applying the two aforementioned methods to process protein sequence, a 3D matrix and a 2D matrix are obtained. These matrices contain rich evolution information about target protein, enabling effective description of protein sequences and further utilization in predicting binding affinity. Importantly, the generation of these two matrices is straightforward and does not require assistance from other tools, making them adaptable to any protein with a sequence and highly efficient for large-scale virtual screening applications.

To represent the features of small molecules, we utilize a graph-based approach. The SMILES of molecule is read by RDKit toolkit (Landrum, 2006), and it is transformed into the corresponding graph. The graph is constructed with atom as node and chemical bond as edge. In order to get the hidden structural information of the molecule, the graph is further processed by the graph neural network, which a hidden feature is obtained. Then the feature is used to predict the affinity with protein.

Model construction

Due to the two processing ways of the protein, there are two networks used for the protein feature extraction. Thus, the whole model involves three entrances and three corresponding networks, which two of them are used for protein feature mining and the other one is used for molecule feature mining. Then, the three obtained hidden features are concatenated to prediction the affinity. The whole model architecture is illustrated as Fig. 4.

Figure 4 In this process, a three-dimensional matrix is used with 3DCNN to extract features, a two-dimensional matrix is processed with 2DCNN to extract features, and a small molecule graph is processed with GNN to extract features.

The obtained features are then fused and passed through a fully connected neural network to predict drug-target affinity values.

Because of the difference of the two processing ways for protein sequence, there are two matrices generated with different shapes. The first processing is utilized with k-mers segmentation with a k of 3 and the shape of the generated matrix OsK is L×L×L. The second processing handle sequence using Cartesian product calculation operation, and it could result in a matrix OsC with a shape of L×L. So, two convolutional neural networks are involved for the protein, where the matrix generated by k-mers segmentation is fed into a 3D convolutional neural network, and another 2D convolutional neural network is used for the matrix generated by Cartesian product calculation.

Since the molecule is represented by graph, a graph neural network (GNN) is utilized for its feature extraction. The model is utilized with five GNN layers and each layer is followed by a batch normalization layer. To find the best GNN model, we tried to establish the GNN model with three different types of GNN layers: GCN (Graph Convolutional Network), GAT (graph attention network), and GIN (Graph Isomorphism Network), and the performance of different types of GNN layers was thoroughly discussed in the experimental section.

The Graph Convolutional Network (GCN) (Kipf & Welling, 2016) is a specialized neural network model designed for processing graph-structured data. It utilizes the topological structure of the graph to perform convolutional operations at each node, updating their representations by aggregating information from neighboring nodes. By leveraging local information propagation and computation, GCN efficiently learns feature representations of nodes in the graph. The calculation of GCN layer is as follows:

(1) H(l+1)=σ(D−12A^D−12H(l)W(l))

where, H(l+1) represents the updated node feature matrix at layer l+1, σ denotes the activation function. D−12 is the diagonal degree matrix of the graph, where each diagonal element is the inverse of the square root of the corresponding node’s degree. A^ is the adjacency matrix of the graph with self-loops added. H(l) is the node feature matrix at layer l. W(l) denotes the weight matrix of the graph convolutional layer at layer l.

GAT (Veličković et al., 2017), short for Graph Attention Network, is a neural network model that introduces attention mechanisms to dynamically attend to important relationships among nodes, thereby facilitating more effective learning of graph-structured data representations. The GAT layer takes the set of nodes in the graph as input and applies a linear transformation to each node using a weight matrix W. For each input node i in the graph, the attention coefficients eij between node i and its first-order neighbors are computed as follows:

(2) eij=a(Wxi,Wxj)

where a represents the attention function. This value signifies the significance of node j with respect to node i. Subsequently, these attention coefficients undergo normalization using the softmax function, αij are the normalized attention coefficients:

(3) αij=softmaxj(eij)=eeij∑k∈N(i)eeik

Afterward, the output features hij for the nodes are computed using the following equation:

(4) hij=σ(∑j∈N(i)αijWxj)

And σ is a non-linear activation function.

GIN (Xu et al., 2018), or Graph Isomorphism Network, is a deep learning model that is specifically designed to handle data in graph structures. Graphs are complex data structures where each node can have multiple neighbors, and the number and position of these neighbors can vary significantly between different graphs. This variability makes it challenging for traditional neural networks to effectively process such structures. However, GIN leverages the graph isomorphism assumption, which states that two graphs are considered isomorphic if one can be transformed into another by relabeling their nodes. By exploiting this property, GIN is able to effectively and efficiently process any graph neural structure data, regardless of its size or complexity. This makes GIN a powerful tool for a wide range of applications, including social network analysis, molecular chemistry, and recommender system. The formula of GIN is as follows:

(5) hv(k)=MLP(k)((1+ϵ(k))∗hv(k−1)+∑u∈N(v)hu(k−1))

Here, hv(k) represents the feature vector of node v in layer k, N(v) represents the set of neighboring nodes of node v, MLP(k) represents a multilayer perceptron, and ε(k) is a learnable bias vector.

Because different molecules have different number of atoms, the molecular graph could be constructed with different number of nodes. So, a global pooling layer is added to ensure that the hidden vectors extracted for different molecules could have the same dimension. Finally, the two hidden features of protein and the hidden feature of molecule are concatenated and put into a fully connected neural network to predict the affinity.

Datasets

Four benchmark datasets were utilized in this experiment, namely Davis (Davis et al., 2011), KIBA (Tang et al., 2014), Metz (Metz et al., 2011), and PDBBind refined (Liu et al., 2015). Among them, Davis and KIBA are two widely used datasets that can provide a more objective evaluation of KC-DTA’s performance. Davis is a medium-sized dataset, while KIBA is larger in scale. In addition, PDBBind refined is a smaller-scale dataset, and we have also employed a dataset, Metz, of a similar scale to Davis to comprehensively assess the performance of our method across datasets of varying sizes. The introduction of the following datasets is as follows:

Davis dataset: The Davis dataset comprises selectivity assays of kinase protein families and associated inhibitors, along with their respective dissociation constant ( Kd) values, which consists of the binding affinities of 68 inhibitors to 442 protein kinases in the human protein kinase group and yields a total of 30,056 binding entities. For Davis dataset, the same measure of SimBoost is used, which the affinities are transformed into log space, defined as:

(6) pKd=−log10(Kd1e9)

KIBA dataset: The KIBA dataset is developed using a technique called KIBA, which integrates Ki, Kd, and IC50 by optimizing consistency among them. The processed KIBA dataset comprises a total of 118,254 binding entities from 229 proteins and 2,111 small molecules.

Metz dataset: The Metz dataset is a significant public resource that aims to facilitate the development of computer-aided drug design. It has gained widespread usage in the fields of machine learning and drug discovery and is regarded as one of the most frequently employed public datasets. In our experiment, the Metz dataset consists of 1,423 small molecules, 170 proteins, and 35,259 binding entries.

PDBBind dataset: PDBBind dataset (Liu et al., 2015; Burley et al., 2017) is sourced from the Protein Data Bank, and it is a valuable resource for predicting the binding affinity of protein-ligand complexes. The dataset consists of three subsets, namely the general set, the refined set, and the core set. The general set consists of complexes with generic mass, while the refined set provides details on complexes with high structural resolution. Although the core set is of the highest quality, the size is very small so that the set could not be used for data mining. The PDBBind refined dataset (v.2015) is involved for the experiments, which the number amounts to 3,047 protein-molecule pairs.

The numbers of proteins, molecules and binding affinity entities and the data sources for each dataset are illustrated in Table 1.

Table 1 The numbers of interactions for each dataset.

Dataset	Compounds	Proteins	Interactions	Sources	
Davis	68	442	30,056	https://github.com/hkmztrk/DeepDTA	
KIBA	2,111	299	118,254	https://github.com/hkmztrk/DeepDTA	
Metz	1,423	1,708	35,259	https://github.com/simonfqy/PADME	
PDBBind (refined)	2,291	1,960	3,047	https://github.com/cansyl/MDeePred	

Results

In order to comprehensively measure the performance of the proposed model, some experiments are carried out with several metrics. In this experiment, we chose PyTorch 2.0.0 (Paszke et al., 2019) to build our model. Our experiments were carried out on a server equipped with 128 GB of memory and running the Ubuntu operating system. To expedite the training process, we utilized two NVIDIA GeForce RTX 4090 graphics cards along with cuda 11.8 (Sanders & Kandrot, 2010).

Metrics

The Concordance Index (CI) is widely used to evaluate the performance of regression models and could be used as an evaluation measure of prediction accuracy. The CI for a set of paired data is equal to the probability that two drug-target pairs with different label values are predicted in the correct order, and a higher CI value indicates a better predictive performance. The calculation of CI is defined as follows:

(7) CI=1N∑y^i>y^jh(yi−yj)

where yi and yj are the predicted values of larger affinity value y^i and smaller affinity value y^j respectively, and N is a normalization constant representing the number of pairs in the correct order, h(x) is the Heaviside step function, defined as:

(8) h(x)={1,x>00.5,x=00,x<0

MSE is the most popular metric for evaluating regression models and measures how close the fitted line is to the actual data points. MSE is defined as the mean of the sum of squares of the differences between the true value and the predicted value. So, a lower value of MSE indicates a better predictive performance. The MSE calculation is defined as:

(9) MSE=1n∑i=1n(y^i−yi)2

where y^i is the predicted value vector, yi is the actual value vector, n is the number of samples, and MSE is the average of the sum of squares of the difference between the predicted value and the actual value. The smaller the MSE value, the higher the efficiency of the regression model. Root mean squared error (RMSE) is the square root of the root mean square error, which is used in the PDBBind dataset in this experiment.

Another metric rm2 (Roy, 2015) has been widely used to validate regression-based quantitative structure-activity relationship (QSAR) models. rm2 is a modified version of the squared correlation coefficient, also known as the coefficient of determination (r2), used to assess the external predictive potential of binding affinity models. If the value of the rm2 index is greater than 0.5 in the test set, the model is acceptable. Where r2 and r02 are the square of the correlation coefficient with and without intercept, respectively. rm2 is the proportion of variables by which the variable predicts the described outcome. The formula is shown in 10, and more details of the formula can be found in previous studies (Pratim Roy et al., 2009; Roy et al., 2013).

(10) rm2=r2⋅(1−r2−r02)

Ablation experiment

In order to obtain the best model for the drug-target affinity prediction, we employed nine combinations for the model construction. For protein, we utilized three approaches of model construction: a single 2D+CNN for the feature extraction on two-dimensional matrix (2D) derived from Cartesian product calculation operation, a single 3D+CNN for the feature extraction on three-dimensional matrix (3D) derived from k-mers analysis, and both 2D+CNN and 3D+CNN involved for the two matrices. For molecule, three types of GNNs are utilized, which involve GCN (Kipf & Welling, 2016), GAT (Veličković et al., 2017), and GIN (Xu et al., 2018). Thus, a total of nine different combinations of model architectures are used for the performance comparison, as presented in Tables 2 and 3. We conducted five-fold cross-validation experiments on Davis and KIBA datasets to evaluate the performance of these nine combinations. The evaluation metrics used were CI, MSE, and rm2.

Table 2 The performances of different combinations on Davis dataset.

Method	Proteins	Compounds	CI (std)	MSE (std)	rm2 (std)	
2D+GCN	2DCNN	GCN	0.885 (0.0013)	0.249 (0.0053)	0.644 (0.0161)	
2D+GAT	2DCNN	GAT	0.884 (0.0033)	0.246 (0.0052)	0.650 (0.0101)	
2D+GIN	2DCNN	GIN	0.885 (0.0033)	0.242 (0.0052)	0.659 (0.0100)	
3D+GCN	3DCNN	GCN	0.883 (0.0017)	0.245 (0.0045)	0.655 (0.0099)	
3D+GAT	3DCNN	GAT	0.886 (0.0020)	0.245 (0.0047)	0.656 (0.0074)	
3D+GIN	3DCNN	GIN	0.880 (0.0010)	0.256 (0.0024)	0.640 (0.0055)	
2D+3D+GCN	2DCNN+3DCNN	GCN	0.884 (0.0050)	0.241 (0.0036)	0.658 (0.0097)	
2D+3D+GAT	2DCNN+3DCNN	GAT	0.886 (0.0037)	0.240 (0.0040)	0.661 (0.0154)	
2D+3D+GIN	2DCNN+3DCNN	GIN	0.888 (0.0040)	0.235 (0.0033)	0.658 (0.0047)	

Table 3 The performances of different combinations on KIBA dataset.

Method	Proteins	Compounds	CI (std)	MSE (std)	rm2 (std)	
2D+GCN	2DCNN	GCN	0.886 (0.0044)	0.146 (0.0035)	0.758 (0.0036)	
2D+GAT	2DCNN	GAT	0.862 (0.0032)	0.178 (0.0044)	0.703 (0.0118)	
2D+GIN	2DCNN	GIN	0.876 (0.0018)	0.160 (0.0016)	0.747 (0.0041)	
3D+GCN	3DCNN	GCN	0.887 (0.0031)	0.146 (0.0014)	0.761 (0.0080)	
3D+GAT	3DCNN	GAT	0.865 (0.0014)	0.176 (0.0033)	0.713 (0.0045)	
3D+GIN	3DCNN	GIN	0.873 (0.0017)	0.162 (0.0027)	0.745 (0.0061)	
2D+3D+GCN	2DCNN+3DCNN	GCN	0.890 (0.0013)	0.143 (0.0018)	0.759 (0.0057)	
2D+3D+GAT	2DCNN+3DCNN	GAT	0.861 (0.0016)	0.180 (0.0023)	0.702 (0.0060)	
2D+3D+GIN	2DCNN+3DCNN	GIN	0.877 (0.0013)	0.159 (0.0014)	0.751 (0.0067)	

From Table 2, it is evident that the 2D+3D representation consistently outperforms using either 2D or 3D alone for the protein representation. Specifically, the MSE of 2D+3D+GCN is 1.6% lower than that of 3D+GCN and 3.2% lower than that of 2D+GCN. The MSE of 2D+3D+GAT is 2.0% lower than that of 3D+GAT and 2.4% lower than that of 2D+GAT. The MSE of 2D+3D+GIN is 8.2% lower than that of 3D+GIN and 2.9% lower than that of 2D+GIN. This can be attributed to the ability of the 2D+3D combination to capture protein sequence information more accurately, thus enhancing accurate prediction. Among the 2D+3D combinations,

From Table 3, the combination of 2D+3D+GCN outperforms in all metrics. In terms of MSE, 2D+3D+GCN is 10.1% lower than 2D+3D+GIN and 20.6% lower than 2D+3D+GAT. Although 2D+3D+GIN exhibits the best performance on the Davis dataset, the margin by which it surpasses 2D+3D+GCN is relatively small. Therefore, considering a comprehensive assessment, we opt for the combination of 2D+3D+GCN for all subsequent experiments.

Performance on Davis and KIBA datasets

For the Davis and KIBA datasets, we employ a five-fold cross-validation method applied in DeepDTA (Öztürk, Özgür & Ozkirimli, 2018). This involves dividing the dataset into six equal parts with one part being reserved for testing and the remaining five partitions used as cross-validation sets. During each fold of cross-validation, one of the cross-validation sets is designated as a validation set for model training while the other four are used for actual training. After training on the training set, the model’s parameters are adjusted based on its performance on the validation set. The best-performing model is then saved, and its parameters are used to evaluate performance on the test set. The performance evaluation uses CI, MSE, and rm2, which is consistent with the measures employed in DeepDTA. Seven benchmarks including KronRLS (Pahikkala et al., 2015), SimBoost (He et al., 2017), DeepCPI (Tsubaki, Tomii & Sese, 2019), DeepDTA (Öztürk, Özgür & Ozkirimli, 2018), GANsDTA (Zhao et al., 2020), MPS2IT-DTI (De Souza, Fernandes & de Melo Barbosa, 2022), WideDTA (Öztürk, Ozkirimli & Özgür, 2019) are involved for the performance evaluation and the results are illustrated in Figs. 5 and 6.

Figure 5 The performances of various methods on Davis dataset.

Figure 6 The performances of various methods on the KIBA dataset.

Figures 5 and 6 present the performance evaluation of different methods on the Davis and KIBA datasets, indicating that KC-DTA outperforms other methods on the datasets with metrics of CI, MSE and rm2. The performance on Davis with MSE and rm2 metrics is slightly greater than the other methods, but when it comes to the KIBA dataset, the performance improvement is significant for the KC-DTA, which the MSE of it could achieve 0.143. Specifically, KC-DTA improves upon the second-ranked WideDTA by 1.71% in CI and 20.1% in MSE, and achieves an impressive 12.4% improvement in rm2 over the second-ranked GANsDTA on KIBA dataset.

Performance on Metz dataset

The same measure with MGraphDTA is utilized for this experiment, which the dataset is randomly divided into five parts with a ratio of 4:1 and the four parts are used as the training set and the last part is used as the test set. To ensure the robustness of the proposed method, we performed the random division three times using different seeds, resulting in three distinct experimental outcomes. The final experimental result was obtained by computing the average of these three outcomes. There are three performance metrics used by the Metz dataset, namely MSE, CI, rm2 and all the results of the other methods come from the relevant articles. The detailed experimental results are shown in Fig. 7.

Figure 7 The performances of different combinations on the Metz dataset.

Figure 7 exhibits that the proposed method could achieve a good performance on the dataset. The performance of KC-DTA with CI and rm2 metrics on the Metz dataset is slightly lower than the top performing method, but the performance of it with MSE could reach a best result, which the MSE is 0.258. Compared with GraphDTA, although the processing of small molecules is the same, KC-DTA’s performance is significantly optimized owing to the use of k-mers segmentation and Cartesian product calculation for protein representation that better reflect the completeness of protein sequence information and thereby enhance the accuracy of drug-target affinity prediction.

Performance on PDBBind dataset

In this experiment, three methods were included for the performance comparison, which are Affinity2Vec (Thafar et al., 2022), MoleculeNet (Mousavian & Masoudi-Nejad, 2014), MDeePred (Rifaioglu et al., 2021) and the proposed method. PDBBind Refined dataset (v.2015 dataset) is utilized in the experiment and a similar processing procedure to the Affinity2Vec (Thafar et al., 2022) is employed. Specifically, the drug-target pairs in the dataset that were published on or before 2011 as the training set, those published on 2012 as the validation set, and those published on or after 2013 as the test set. The dataset comprised a total of 2,188, 312, and 547 pieces of data for the training, validation, and test sets, respectively. In MoleculeNet, four methods including deep neural network (DNN), random forest (RF), grid featurization (GridF) and extended connection fingerprint (ECFP) were involved for the processing of molecule and protein. And Affinity2Vec includes three different ways, which are embedding (Embed), Protein score (Pscore) and Hybrid Protein Identification Scoring Algorithm (Hybrid). The experimental results are illustrated in Fig. 8.

Figure 8 The performances of different combinations on the PDBBind dataset.

Figure 8 indicates that the proposed KC-DTA method is generally comparable to MDeePred in terms of RMSE but slightly inferior to Affinity2Vec. Specifically, KC-DTA exhibits a lower RMSE of approximately 3.8% compared to MDeePred, while CI shows a slight improvement compared to most methods.

Discussion

KC-DTA processes protein sequences using k-mers and Cartesian product, avoiding truncation of protein sequences and preserving the integrity of important information. Additionally, KC-DTA does not rely on complex algorithmic tools, allowing it to quickly transform protein sequences. This simple, fast, and effective method can be easily applied to large batches of protein and small molecule activity predictions. From experimental results, KC-DTA performs better overall on the Davis, Metz, and KIBA datasets compared to other methods. On the PDBBind dataset, our approach slightly lags behind Affinity2Vec. The PDBBind dataset has only 2,188 training samples, and these limited data may not be sufficient for the model to fully fit, thus preventing it from realizing its full potential. In summary, KC-DTA excels on medium to large datasets and demonstrates relative stability on small datasets, making it a promising deep learning approach.

In addition, our method has not yet been applied in practical cases, making it difficult to verify its performance on larger datasets or in scenarios with more severe situation. In the future, we will attempt experiments with a broader array of databases to ensure that the model can adapt to a wide variety of situations, thereby enhancing the robustness of KC-DTA.

Conclusions

The accurate prediction of drug-target affinity is essential for the development of effective pharmaceuticals. While identifying whether a drug binds to its target is important, it is equally crucial to predict the precise value of the drug-target affinity. In this study, we propose a novel sequence-based method for predicting drug-target affinity using protein sequence and drug SMILES. By leveraging k-mers analysis operation and Cartesian product calculation methods, we convert protein sequences into two different matrices, which could better capture the contact among residues and evolutionary information in sequence. The proposed method is easily implementable and applicable to large-scale virtual screening tasks.

Additional Information and Declarations

Competing Interests

Author Contributions

Data Availability

The authors declare that they have no competing interests.

Mingjian Jiang analyzed the data, prepared figures and/or tables, authored or reviewed drafts of the article, and approved the final draft.

Yunchang Shao conceived and designed the experiments, performed the experiments, prepared figures and/or tables, and approved the final draft.

Yuanyuan Zhang analyzed the data, authored or reviewed drafts of the article, and approved the final draft.

Wei Zhou analyzed the data, authored or reviewed drafts of the article, and approved the final draft.

Shunpeng Pang analyzed the data, authored or reviewed drafts of the article, and approved the final draft.

The following information was supplied regarding data availability:

The data and code are available at GitHub:

- https://github.com/hkmztrk/DeepDTA, Davis and KIBA, Bogazici University;

- https://github.com/simonfqy/PADME, Qingyuan Feng, Metz, Sun Yat-sen University;

- https://github.com/cansyl/MDeePred, PDBBind, Middle East Technical University.

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
