# Peer review of "A deep learning method for drug-target affinity prediction based on sequence interaction information mining"

_PeerJ, doi:10.7717/peerj.16625_

## Round 0.1 · original submission · Major Revisions

Dear Dr. Jiang and colleagues:

Thanks for submitting your manuscript to PeerJ. I have now received four independent reviews of your work, and as you will see, the reviewers raised some concerns about the research. Despite this, these reviewers are very optimistic about your work and the potential impact it will have on research studying sequence-based drug target design. Thus, I encourage you to revise your manuscript, accordingly, considering all the concerns raised by the four reviewers.

Please work to make your revision clearer in its presentation and content. The authors provide numerous examples where clarity is needed (or revised sections with more background). The methods should also be clear, concise, and repeatable. Please ensure this, and make sure all relevant information and references are provided. Also, elaborate on the discussion of your findings, placing them within a broad and inclusive body of work by the field (though terminology should be carefully used to make your work interesting to a broader audience). Some of the figures need revision.

Please fix all the identified grammatical issues.

You may leave the Results and Discussion merged.

Therefore, I am recommending that you revise your manuscript, accordingly, considering all the issues raised by the reviewers.

I look forward to seeing your revision, and thanks again for submitting your work to PeerJ.

Good luck with your revision,

-joe

**Language Note:** The Academic Editor has identified that the English language must be improved. PeerJ can provide language editing services - please contact us at copyediting@peerj.com for pricing (be sure to provide your manuscript number and title). Alternatively, you should make your own arrangements to improve the language quality and provide details in your response letter. – PeerJ Staff

Reviewer 1 ·

Basic reporting

The Result and Discussion section to be separated as two independent sections as per PeerJ Journal policy
Figure legends to be detailed
References for example in line 58 is confusing WideDTA WideDTA ¨Ozt ¨urk et al. (2019)..similar usage is seen throughout the journal like Davis Davis et al. (2011), Metz Metz et al. (2011)
Line 148: "For molecule, we use graph to represent it.." Re frame this sentence

Experimental design

no

Validity of the findings

no

Reviewer 2 ·

Basic reporting

Give a dependent section of comprehensive discussion including limitations that need to be explored in the future.

Experimental design

The introduction could cite more recent literature to provide more background on recent advancements and limitations in deep learning methods for drug-target affinity prediction.

Validity of the findings

The model performs worse on the PDBBind dataset compared to Davis, KIBA, and Metz. Could there are more hyperparameter tuning and architecture modifications be explored to improve generalization to this dataset?

·

Basic reporting

a. Writing is pretty clear.
b. References are rich, and most of the related methods are provided in the introduction.
c. Structure is okay. I suggest elaborate methods before the dataset. However, some figures are ambiguous.
- In Figure 2, the 3D matrix, the count number of each 3-mers should be represented by a unit cube instead of a face.
- In Figure 4, do 3DCNN, 2DCNN, and GNN blocks mean the whole NN model, or just one layer?
- Table 3, and Table 4 should be placed above the references.

Experimental design

a. In table 2, I don't see significant differences between of those nine combinations. In general, models with more complexity (more parameters like 2D+3D+GIN here) can squeeze the bias but always introduce more variance. Instead of just calculating the metrics once, cross-validation is always used to select the most robust and generalizable model.

b. Authors compared their KC-DTA method with plenty of benchmarks on different datasets. Among them, why hasn't the RM metric of WideDTA been reported? Also, Affnity2Vec with Embed model FV seems to be better than your method in Table 4. Some explanations are needed.

Validity of the findings

This paper shows an interesting framework, called KC-DTA, for drug-target affinity prediction based on deep learning methods like CNN and GNN and evaluates the method on various datasets by comparing with different benchmarks. In general, it tells a complete story, and the code repo is solid.

However, the novelty is limited because their model structure follows the same design as Graph-DTA paper. For example, the performance of their model is very similar to GraphDTA with CNN+GAT-GCN in table 3, as well as their model structure. Although they consider different encoding methods and make some modifications in the model architecture, a detailed comparison with Graph-DTA needs to be elaborated.

Nguyen, T., Le, H., Quinn, T. P., Nguyen, T., Le, T. D., and Venkatesh, S. (2021). Graphdta: Predicting
drug–target binding affinity with graph neural networks. Bioinformatics, 37(8):1140–1147.

·

Basic reporting

(1) Line 46-70 : Within the introduction it is helpful to see the different approaches for affinity predictions, and to understand the capabilities and specifications of other methods. What might be beneficial here is to add a few sentences regarding the gaps that this work is attempting to address from a few of the methods described as part of the literature review.

(2) Line 33: Pertaining to raw data, the GitHub link is very informative and useful. Is there a possible way for the authors to identify within the readme, where the codebase pertaining to the data transformations employed for the protein structures, and the code used to get the SMILES encoding is located? And whether it is easily usable by the community? This is a pivotal contribution of this work and it would be very useful if there were usable examples reproducible by readers.

(3) Line 56: Possible references for SMILES encoding and a one line explanation might be beneficial .

(4) Table 3 and 4 appear in the middle of the reference section in my copy of the manuscript, would be helpful is the placement of the tables was next to the relevant section.

(5) Line 311-313: "The decrease of the performance on the PDBBind Refined dataset because the number of data entries in it is relatively small, which the number of data entries of Davis, Metz, and KIBA datasets are 20,037, 28,207, and 78,836, respectively, compared to the 2,188 data entries in the PDBBind Refined dataset." - This point could possibly be highlighted with slight restructuring and rephrasing. Additionally, a comment on why the current method is increasingly successful in presence of more data might be helpful.

Experimental design

(1) Line 109: It would be extremely beneficial to have some added understanding around why two different types of protein representations were enabled and utilized by the methodology. Informing the reader at the outset of the benefits k-mer segmentation representation and the Cartesian product enabling all possible residue combinations, possibly in the context of how affinity might be impacted or gleaned from these will be helpful in further motivating the methodology.

(2) Line 154: If it is feasible, the authors can add detail around the processing power required to build/train and employ these neural network combinations for affinity prediction. This would be beneficial for understanding the scalability of this method for further use.

(3) Line 247: From my understanding, table 2 is showing for all three data modalities (k-mer protein, residue combination protein and molecule structure) the use of different approaches for feature extraction and modeling. This section can be rephrased to reflect this better. Possibly with the help of either expanding the table to have 3 columns (each for the different data and the method used), or with the help of restructuring the text a little bit.

(4) Line 257: Could the authors possibly elucidate why just one of the datasets was utilized in the ablation experiment? Is it possible to add data for another dataset modeling and prediction as supplementary. Considering the outcomes did not show large differences (<5% change), might be helpful to confirm the selection by using another dataset and hopefully arriving at the same combination of methods outperforming the others.

(5) Could table 3 and 4 also be figures similar to 6 and 5? It would be helpful to see the performance within similar or the same figure.

Validity of the findings

(1) Line 258- 315 : It would be helpful to have a few lines highlighting to the readers the reasons for the selection of the "comparison/benchmarking" methods for each dataset. Were different benchmarks utilized due to the structure of the data in each dataset?

(2) In a similar vein as above, is it feasible to train on a dataset, and do an external validation on another? Completely understandable if this is not feasible, but metrics around prediction of such a validation, where the training dataset is completely separately sourced from the external testing data would be further evidence of the power of this methodology.

(3) It would be beneficial to the work if there was some more information around how the proposed methodology is beneficial for large scale screening tasks, and possibly some comparisons around how scalable the data transformations involved are. A few lines comparing the data transformations specifically to other method benchmarks and providing further evidence of the benefits of the stand alone data transformations, which don't have many dependencies will be helpful to readers and further motivate and highlight the merits of this work.

Additional comments

I thank the authors for presenting this work on drug-target affinity prediction. The manuscript is well written and addresses a need for a method that can perform this prediction in a straightforward manner at scale. The structure of the framework and analysis was well explained, as were the data sources used as benchmarks. I have a few thoughts outlined in sections above that can possibly help highlight the value and contribution of this method further. Specifically the suggestions above refer to slight restructuring of parts of the text to highlight the points being made, and adding some clarifying text to address some of the decisions made during the construction and employment of this method. Also a few suggestions are included on how the work can be made more available to the research community as well as allowing for reproducibility.

---

## Round 0.2 · accepted · Accept

Dear Dr. Jiang and colleagues:

Thanks for revising your manuscript based on the concerns raised by the reviewers. I now believe that your manuscript is suitable for publication. Congratulations! I look forward to seeing this work in print, and I anticipate it being an important resource for groups studying sequence-based drug target design. Thanks again for choosing PeerJ to publish such important work.

Best,

-joe

Reviewer 2 ·

Basic reporting

no comment

Experimental design

no comment

Validity of the findings

no comment

Additional comments

no comment

·

Basic reporting

No comment

Experimental design

No comment

Validity of the findings

No comment

Additional comments

The authors have addressed all my previous comments with clarity and substantial changes to the manuscript. I feel all my previous notes have been utilized to provide a stronger manuscript.